**Data Availability Statement:** Restrictions by the data custodians mean that the datasets are not

# Cardiopulmonary, metabolic, and perceptual responses during exercise in Myalgic Encephalomyelitis/Chronic Fatigue Syndrome (ME/CFS): A Multi-site Clinical Assessment of ME/CFS (MCAM) sub-study

Dane B. Cook[1,2]*, Stephanie VanRiper[1,2], Ryan J. Dougherty[3], Jacob B. Lindheimer[1,2,4], Michael J. Falvo[5,6], Yang Chen[7], Jin-Mann S. Lin[7], Elizabeth R. Unger[7], The MCAM Study Group[¶]

1 William S. Middleton Memorial Veterans Hospital, Madison, WI, United States of America, 2 Department of Kinesiology, University of Wisconsin-Madison, Madison, WI, United States of America, 3 Department of Epidemiology, Johns Hopkins Bloomberg School of Public Health, Baltimore, Maryland, United States of America, 4 Department of Medicine, University of Wisconsin-Madison, Madison, WI, United States of America, 5 War Related Illness and Injury Study Center, Department of Veterans Affairs New Jersey Health Care System, East Orange, NJ, United States of America, 6 New Jersey Medical School, Rutgers Biomedical and Health Sciences, Newark, NJ, United States of America, 7 Centers for Disease Control and Prevention/ Division of High-Consequence Pathogens and Pathology/Chronic Viral Diseases Branch, Atlanta, Georgia, United States of America

¶ Membership of the Multi-site Clinical Assessment of ME/CFS (MCAM) Study Group is provided in the acknowledgments.
* dane.cook@wisc.edu

## Abstract

### Background

Cardiopulmonary exercise testing has demonstrated clinical utility in myalgic encephalomyelitis/chronic fatigue syndrome (ME/CFS). However, to what extent exercise responses are independent of, or confounded by, aerobic fitness remains unclear.

### Purpose

To characterize and compare exercise responses in ME/CFS and controls with and without matching for aerobic fitness.

### Methods

As part of the Multi-site Clinical Assessment of ME/CFS (MCAM) study, 403 participants (n = 214 ME/CFS; n = 189 controls), across six ME/CFS clinics, completed ramped cycle ergometry to volitional exhaustion. Metabolic, heart rate (HR), and ratings of perceived exertion (RPE) were measured. Ventilatory equivalent ($\dot{V}E/\dot{V}O_2$, $\dot{V}E/\dot{V}CO_2$), metrics of ventilatory efficiency, and chronotropic incompetence (CI) were calculated. Exercise variables were compared using Hedges' $g$ effect size with 95% confidence intervals. Differences in cardiopulmonary and perceptual features during exercise were analyzed using linear mixed

publicly available or able to be provided by the authors. Researchers wanting to access the datasets used in this study should email CDC's ME/CFS Program (cfs@cdc.gov) and discuss next steps for the data request. The ME/CFS program data review committee will grant the access after the review and the data use agreement is finalized. Examples of SPSS syntax for key analyses will be available per request. No special access to the data was granted to the authors which would not be available to other researchers.

**Funding:** The study was funded by the Centers for Disease Control and Prevention. Centers for Disease Control Contract AAD3581 to the University of Wisconsin - Madison. DBC. Sponsor (CDC study team) designed, collected the data, and helped prepare the manuscript.

**Competing interests:** No authors have competing interests.

effects models with repeated measures for relative exercise intensity (20–100% peak $\dot{V}O_2$). Subgroup analyses were conducted for 198 participants (99 ME/CFS; 99 controls) matched for age (±5 years) and peak $\dot{V}O_2$ (~1 ml/kg/min$^{-1}$).

## Results

Ninety percent of tests (n = 194 ME/CFS, n = 169 controls) met standard criteria for peak effort. ME/CFS responses during exercise (20–100% peak $\dot{V}O_2$) were significantly lower for ventilation, breathing frequency, HR, measures of efficiency, and CI and significantly higher for $\dot{V}E/\dot{V}O_2$, $\dot{V}E/\dot{V}CO_2$ and RPE ($p<0.05_{adjusted}$). For the fitness-matched subgroup, differences remained for breathing frequency, $\dot{V}E/\dot{V}O_2$, $\dot{V}E/\dot{V}CO_2$, and RPE ($p<0.05_{adjusted}$), and higher tidal volumes were identified for ME/CFS ($p<0.05_{adjusted}$). Exercise responses at the gas exchange threshold, peak, and for measures of ventilatory efficiency (e.g., $\dot{V}E/\dot{V}CO_{2nadir}$) were generally reflective of those seen throughout exercise (i.e., 20–100%).

## Conclusion

Compared to fitness-matched controls, cardiopulmonary responses to exercise in ME/CFS are characterized by inefficient exercise ventilation and augmented perception of effort. These data highlight the importance of distinguishing confounding fitness effects to identify responses that may be more specifically associated with ME/CFS.

## Introduction

Exercise testing is a valuable methodologic and clinical tool in myalgic encephalomyelitis/chronic fatigue syndrome (ME/CFS). Maximal and submaximal exercise protocols have been designed to test and predict cardiopulmonary responses to acute effort [1–6], ascertain exercise tolerance and disability status [7–10], guide exercise prescription [11, 12], and challenge physiological systems (e.g. immune, autonomic & central nervous systems) to gain insights into ME/CFS pathophysiology and the elicited post-exertional malaise (PEM) [8, 13–24].

To date, numerous exercise studies have reported lower aerobic fitness, with gas exchange thresholds (GET) occurring at a lower percentage of peak oxygen consumption and lower peak aerobic capacity in ME/CFS compared to controls [1, 5, 6, 25–29], although not all studies have reported aerobic fitness differences [4, 30–33]. Meta-analytic methods [34] have been used to determine the pooled effect size difference for peak oxygen consumption. Although quality of the included studies varied greatly (i.e., 16 of the 32 studies did not use criteria for determining peak effort), and the results were heterogenous, the meta-analysis reported that the mean effect difference of 5.2 ml/kg/min lower in ME/CFS compared to controls was moderate and clinically meaningful. More recently, serial exercise tests conducted 24 hours apart (i.e., two-day cardiopulmonary exercise testing (CPET)) have been utilized in ME/CFS to test the ability of the cardiopulmonary system to reproduce physiological performance [35]. These studies have reported an earlier GET and, less consistently, lower peak oxygen consumption [10, 28, 36–39] on the second day of testing compared to the first, suggesting a dysfunctional cardiopulmonary response when the system is serially challenged.

With notable exceptions, few exercise-capacity studies in ME/CFS have reported in-depth results beyond GET and peak capacities. When ventilatory and metabolic responses are

included, studies have generally reported these measures to be lower throughout exercise, suggestive of an inefficient cardiopulmonary system [6, 28, 40]. Heart rate responses to exercise have also been the focus of several studies [5, 6, 41]. Reports of lower HR with increasing exercise intensity compared to controls [6, 41] and an inability to reach 85% of HR maximum [5] suggest that chronotropic incompetence could partly explain exercise intolerance in ME/CFS. However, these metabolic and HR differences during maximal exercise were not replicated when ME/CFS patients were matched to controls on peak aerobic fitness [2].

The value of exercise testing in ME/CFS is clear, yet important questions remain. A critical question is to what extent "abnormal" exercise responses in ME/CFS are disease specific or are secondary consequences of either low fitness, failure to reach peak effort, and/or comparisons to more fit controls. Therefore, applying standardized peak effort criteria and expressing exercise data relative to peak capacity (i.e., statistically controlling for fitness and exercise time differences) are critical for standardizing group comparisons and controlling for potential confounding effects of aerobic fitness. A more rigorous approach would be to match participants based on their exercise capacity [2].

Our understanding of cardiopulmonary and metabolic function in ME/CFS is also limited by the reporting of only basic variables (e.g., peak oxygen consumption, work rate at anaerobic threshold) derived from exercise testing. Alternative indices that yield additional clinical insight include, but are not limited to, variables that can be calculated to estimate ventilatory efficiency and HR performance such as the oxygen uptake efficiency slope (OUES), HR and metabolic indexes, and oxygen consumption trajectories. This study was conducted to: 1) compare exercise capacity of those with ME/CFS and otherwise healthy controls within the Multi-site Clinical Assessment of Chronic Fatigue Syndrome (MCAM) cohort, 2) to compare cardiopulmonary, metabolic, and perceptual responses to exercise in those with ME/CFS and otherwise healthy controls, and 3) determine the role of aerobic fitness on the exercise response, efficiency, and HR variables of interest.

## Methods

This study was conducted as part of the Centers for Disease Control and Prevention (CDC) MCAM study [42]. This multi-site study enrolled participants from seven specialty clinics in the United States based on expert clinician diagnoses and characterized with standardized assessment tools for illness domains. The major objectives of the MCAM study were to: 1) measure illness domains of ME/CFS and to evaluate patient heterogeneity; 2) describe illness course and the performance of the chosen psychometric instruments; 3) describe medications, laboratory tests, and management tools that are being used by expert clinicians during ME/CFS care; 4) collect biospecimens including saliva samples for cortisol awakening response profiles; and 5) test which measures best distinguish ME/CFS from comparison groups and test for subgroups. Because exercise capacity is an important illness domain, an MCAM sub-study was developed to conduct cardiopulmonary exercise testing to determine "exercise tolerance" and the relationship between exercise-relevant data and other aspects of the study (e.g., symptom severity, duration of illness).

### Recruitment

**Exercise sub-study.** Participants for the exercise sub-study were recruited with separate informed consent from participants in the parent MCAM study and were enrolled from six of the seven participating ME/CFS specialty clinics. The sites included five clinics across five states (CA, NC, NJ, NV, UT) that were coordinated through the *Open Medicine Institute*

*(OMI) Consortium*, Mountain View, CA and one clinic with two testing sites coordinated through the *Institute for Neuro Immune Medicine* (INIM) (FL).

**MCAM (parent study).** As study participants were enrolled in the parent MCAM study, inclusion and exclusion criteria and baseline parent study protocol (including questionnaires and measures) have been described [42]. In brief, medical records were reviewed by clinic staff or study coordinators and screening was conducted by telephone. ME/CFS participants were excluded from the study if illness onset occurred after 62 years of age, they had human immunodeficiency virus infection, were currently pregnant, had dementia, or the participants could not read English at an eighth-grade level. Healthy control participants could not be younger than 18 or older than 70 years of age, self-reported good health, had no history of ME/CFS, and no other active illnesses. After the baseline year, healthy controls were enrolled, matched to a subset of participants with ME/CFS on sex and age (± 5 years). No exclusions were made based on medications used except those indicative of the presence of heart disease (i.e., unsafe for maximal exercise). The study was approved by the Institutional Review Boards of the CDC, OMI [covering Open Medicine Clinic (CA), Hunter-Hopkins Clinic (NC), Richard Podell Clinic (NJ), Bateman Horne Center (UT), and Sierra Internal Medicine (NV)], Mount Sinai Beth Israel (NY), and Nova Southeastern University (INIM clinic, FL).

## Site training

Each site had an exercise specialist who was trained to deliver the exercise testing protocol. Standard operating procedures were delivered to each site and training of site investigators occurred by phone (e.g., to discuss delivery of the protocol) and in-person site visits by the principal investigator of the specialty clinic. Each site practiced delivering the protocol prior to testing the first participant. Calibration tests and cardiopulmonary and work rate outputs from the metabolic testing units were inspected by an independent investigator (DBC) that was not involved in the testing of participants. Upon satisfactory data (i.e., calibration parameters, confirmation of heart rate and work rate outputs, metabolic readouts) the sites were cleared for participant testing.

## Pre-exercise testing

The testing was performed under controlled environmental conditions (20–24˚C and 40–60% relative humidity). Participants were instructed to abstain from smoking for 2 hours, ingesting caffeine or food for 4 hours, and exercising for 24 hours before testing. Compliance with these instructions were confirmed via self-report of the participant prior to testing. Participants were instrumented for monitoring of HR (12-lead electrocardiography (ECG)) and metabolic responses to exercise and a pre-test ECG was conducted to ensure it was safe to initiate exercise testing, and to obtain an initial resting HR measure. For this measurement, participants were asked to remain quiet with eyes closed, arms to the sides, in a restful supine position for 4 minutes.

## Exercise testing

Exercise testing consisted of ramped cycle ergometry to volitional exhaustion. Participants were given one-to two-minutes to acclimate to the instrumentation (i.e., breathing while wearing the facemask) while seated on the cycle ergometer. This was followed by a three-minute, unloaded warm-up. Exercise testing began at 0 Watts and work intensity was increased linearly by 5 Watts every 20 seconds (15 Watts/min) until volitional exhaustion or a point when the prescribed pedal rate could not be maintained. Participants were instructed to maintain a

pedaling cadence of 60–70 revolutions per minute and were verbally encouraged to continue pedaling as long as possible.

Oxygen consumption ($\dot{V}O_2$), carbon dioxide production ($\dot{V}CO_2$), ventilation ($\dot{V}E$), tidal volume ($V_T$), breathing frequency ($f_R$), HR, and work rate measures were obtained during exercise using a metabolic cart and a 2-way non-rebreathing valve attached to an oronasal mask (Hans-Rudolph, Kansas City, MO). The flowmeter was calibrated prior to each exercise test by making multiple comparisons to a three-liter piston syringe. Oxygen and carbon dioxide sensors wercalibrated by the presentation of known gas concentrations. Lactate was measured from capillary blood via a finger-stick and a lactate analyzer at 6 timepoints: rest, minute-2 of exercise, peak exercise and at 3, 6 and 9-minutes post-exercise. Ratings of perceived exertion (RPE) during exercise were measured every two-minutes during exercise using the Borg 6–20 category scale [43] following standard instructional sets. The GET was determined using the V-slope method as described by Sue et al. [44]. Two independent assessors (SVP & RJD) determined the V-slopes. For each participant, breath-by-breath $V\dot{C}O2$ was plotted against $V\dot{O}2$ to visually identify the tangential breakpoint in the $V\dot{C}O2-V\dot{O}2$ relationship. A 20-sec average around this point (10 sec before and after) denoted the GET. Inter-rater differences in CPET parameters (i.e., non-identical 20 second averages) at the time of the ventilatory anaerobic threshold were flagged and adjudicated by the supervising investigator (DBC). Peak effort was determined based on meeting at least 2 of the following criteria: 1) respiratory exchange ratio $\geq 1.1$, 2) achievement of $\geq 85\%$ of age-predicted maximum HR, 3) RPE $\geq 17$, and 4) a change in $\dot{V}O_2$ of $\leq 150$ ml with an increase in work.

From the directly collected measures (i.e., $\dot{V}O_2$, $\dot{V}CO_2$, $\dot{V}E$, HR and Watts) we derived several indices that are indirectly representative of oxygen delivery and ventilatory efficiency. These included ventilatory equivalents of carbon dioxide ($\dot{V}E/\dot{V}CO_2$) and oxygen ($\dot{V}E/\dot{V}O_2$), oxygen pulse ($\dot{V}O_2/HR$), oxygen uptake to work rate ($\dot{V}O_2/WR$) relationship, and the oxygen uptake efficiency slope (OUES). We expressed OUES as the slope of the relationship between $\dot{V}O_2$ (ml/min) and $\dot{V}E$ (L/min) as described by Baba [45] using the following equation:

$$[VO_2 = a \log \dot{V}E + b] \; where \; a = OUES \; and \; b = y-intercept$$

We also assessed several indices of chronotropic incompetence as described in Brubaker et al. [46]. These assessments included whether a participant achieved $\geq 85\%$ of age-predicted maximal HR (APMHR), $\geq 80\%$ of adjusted heart rate reserve (HRR/APMHR–HR$_{rest}$), and calculation of the chronotropic index (CTI) based on estimated HR stages. For the CTI we used the following equation:

$$Estimated \; HR_{stage} = ([220-age-HR_{rest}] \; X \; [(METS_{stage}-1)/(METS_{peak}-1)] + HR_{rest}).$$

Heart rate stages represent estimated (see above formula) and measured heart rates at relative exercise intensities. The CTI is calculated by dividing (measured HR$_{stage}$ / estimated HR$_{stage}$).

## Data processing

Raw exercise data were inspected independently by investigators (RJD & SVR) who were not involved in testing and who were blind to clinical status of participants. Inspection included verification of adherence to established protocols and system calibrations, identification of data artifacts (i.e., non-physiological, missing, or erratic data) that could interfere with interpretation, and determination of whether peak criteria were met. Discrepancies with data interpretation were reported to and resolved in consultation with the supervising investigator (DBC).

From the raw data set, a reduced data set was created for determining peak effort and calculating variables of interest (e.g., OUES, V-slope). This entailed creating 20-second averages for the breath-by-breath data, identifying, and documenting problematic data (e.g., missing or erratic HR data), and calculating the variables of interest. For the 20-sec averages, time was first established similar to the method used by Robergs et al. [47]. This process identified the central time value of each 20-second interval beginning at the identified peak oxygen consumption value and descending in time to the warm-up period. These data were then used to calculate relative exercise intensities (i.e., 0%, 20%, 40%, 60%, 80% and 100% of peak $\dot{V}O_2$) for each participant. This was accomplished by calculating a linear model of $\dot{V}O_2$ predicted by Time (Mean $R^2$ adjusted of all models = 0.918, SD = 0.115) for each exercise test to estimate 95% confidence intervals of Time during which 0%, 20%, 40%, 60%, 80%, and 100% of peak $\dot{V}O_2$ occurred.

## Demographic and functional characteristics of study participants

Demographic data, diagnosis of co-morbid conditions, duration of illness, and questionnaire assessment of symptoms and function were obtained from MCAM records, either baseline (enrollment) data or the most recent clinic visit.

## Statistical analyses

Statistical analyses were conducted using SPSS for Windows (version 26.0.1; SPSS, Chicago, IL) with the exception of the standardized effect size calculations which were calculated using Microsoft Excel as the mean difference between groups divided by the pooled SD, with a Hedges g correction applied to adjust for sample bias. Subject characteristics, measures at the $V_T$, OUES, and peak exercise variables were compared using Hedges' $g$ effect size with 95% confidence intervals [48] with $\alpha = 0.05$. Normality of the repeated measures data was determined by examining skewness, kurtosis, Q-Q plots, and the Shapiro-Wilk test. When non-normal, data were normalized using a two-step approach as described by Templeton [49]. This process first transforms the data by percentile rank. The second step applies an inverse-normal transformation of the percentile rank values. Levene's Test was applied to examine the equality of variances between groups. Missing data for group comparisons were imputed using the Multiple Imputation by Chained Equations method [50] if $\leq$ 15% of the data were missing, otherwise they were handled using listwise deletion.

Differences in cardiopulmonary and perceptual features during exercise including $\dot{V}E$, $f_R$, $V_T$, $\dot{V}E/\dot{V}O_2$, $\dot{V}E/\dot{V}CO_2$, HR, $O_2$ pulse trajectory, CTI and RPE were analyzed using linear mixed effects models with repeated measures for relative exercise intensity. For the mixed effects models, we chose the autoregressive heterogenous covariance structure because proximal data (e.g., 20% and 40%) were more strongly correlated than distal data (e.g., 20% and 100%) and because the Levene's Test revealed unequal variances between groups for several outcome variables. Fixed Effects included Group, Time, Age, and Group*Time and the intercept was included as a Random Effect. For these analyses, both the Group Main Effect and the Group-by-Time interaction were of interest. Only complete exercise tests that met criteria for peak effort were included for analysis and data were expressed relative to peak oxygen consumption to statistically control for differences in fitness and exercise time (detailed above). To more definitively determine the effect of aerobic fitness on the outcomes of interest, we performed the same set of analyses described above on a subgroup of 198 participants (n = 99 ME/CFS; n = 99 controls) matched for peak $\dot{V}O_2$ (± 1 ml/kg/min) and age (± 5 years). Although we did not specifically match based on sex, only 11 pairs (see Results) were not sex

matched. Analysis of the $VO_2$, age and sex-matched subgroup did not substantially alter any of the effect size differences nor the statistical significance of any of the analyses. Further, there were no significant alterations to the results when controlling for race. We also conducted our analyses excluding for the small percentage of participants taking cardiovascular acting drugs (See Table 1) and results were not substantially changed (**See S1 and S2 Data**). This includes resting measures of HR, SBP, and DBP, exercise measures at the GET and peak, and the dynamic responses to exercise. Alpha was set at 0.05 and Holm-Bonferroni Sequential Method was applied to adjust for multiple comparisons [51]. Missing data for these analyses were imputed using the Multiple Imputation by Chained Equations method [50] if $\leq 15\%$ of the data were missing, otherwise they were handled using pairwise deletion.

**Table 1. Demographic and baseline data\* for ME/CFS patients and controls.**

| | Overall Exercise Study Sample | | | Fitness-Matched Subgroup | | |
|---|---|---|---|---|---|---|
| | **ME/CFS (n = 178)** | **Controls (n = 169)** | **ES (CI) or Chi-Square *p*-value** | **ME/CFS (n = 99)** | **Controls (n = 99)** | **ES (CI) or Chi-Square *p*-value** |
| % Female | 65 | 68 | *p* = 0.50 | 61 | 70 | *p* = 0.18 |
| Age (yrs) | 49.4 (13.2) | 42.5 (14.0) | 0.51\*\* (.29 to .72) | 47.3 (13.2) | 47.1 (12.7) | 0.02 (-0.38 to 0.41) |
| Height (m) | 1.7 (0.1) | 1.7 (0.09) | 0.0 (-0.21 to 0.21) | 1.7 (0.09) | 1.7 (0.08) | 0.35 (-.05 to 0.75) |
| Weight (kgs) | 78.5 (18.7) | 73.0 (16.0) | 0.32\*\* (0.10 to 0.53) | 77.4 (16.5) | 76.0 (16.6) | 0.08 (-.31 to 0.48) |
| Education % College Graduate# | 42 | 37 | *p* = 0.41 | 37 | 37 | *p* = 0.49 |
| Smoking Status % Yes | 2.8 | 2.9 | *p* = 0.81 | 2.0 | 4.0 | *p* = 0.35 |
| Race %White## | 94 | 59 | *p* = 0.000 | 97 | 57 | *p* = 0.000 |
| % Comorbid FM### | 43.6 | 6.5 | *p* = 0.000 | 38.9 | 6.7 | *p* = 0.000 |
| % Comorbid IBS### | 35.5 | 9.7 | *p* = 0.000 | 34.7 | 11.1 | *p* = 0.000 |
| % Comorbid Migraine### | 46.5 | 16.1 | *p* = 0.000 | 47.4 | 18.9 | *p* = 0.000 |
| % ACE Inhibitor╫ | 4.6 | 1.9 | *p* = 0.18 | 3.1 | 1.1 | *p* = 0.34 |
| % AR Blocker╫ | 2.9 | 0 | *p* = 0.03 | 4.2 | 0 | *p* = 0.05 |
| % Beta Blocker╫ | 6.4 | 0 | *p* = 0.001 | 6.3 | 0 | *p* = 0.02 |
| % CA2 Inhibitor╫ | 6.4 | 3.9 | *p* = 0.31 | 7.3 | 3.3 | *p* = 0.23 |
| BMI (kg/m$^2$) | 27.3 (6.9) | 26.0 (5.1) | 0.21\*\* (0.00 to 0.42) | 26.7 (5.6) | 27.2 (5.2) | -.09 (-0.49 to 0.30) |
| Resting HR (bpm) | 67.9 (11.6) | 62.2 (10.0) | 0.53\*\* (0.31 to 0.74) | 68.7 (11.3) | 63.5 (10.6) | 0.47\*\* (.19 to 0.76) |
| Resting SBP (mmHg) | 121.8 (14.0) | 121.5 (15.8) | 0.02 (-0.19 to 0.23) | 120.5 (13.5) | 120.5 (15.8) | 0.00 (-0.21 to 0.21) |
| Resting DBP (mmHg) | 79.6 (9.8) | 76.7 (10.6) | 0.28\*\* (0.07 to 0.50) | 79.7 (9.5) | 76.6 (9.9) | 0.32\*\* (0.04 to 0.60) |
| Physical Function\*\*\* | 40.7 (5.3) | 59.0 (6.5) | -3.10\*\* (-3.42 to -2.78) | 41.3 (5.7) | 57.6 (6.9) | -2.58\*\* (-2.96 to -2.20) |
| IPAQ Total (min/week) | 46.1 (79.5) | 106.7 (103.7) | -0.66\*\* (-0.89 to -0.43) | 44.8 (78.0) | 109.7 (113.0) | -0.67\*\* (-0.98 to -0.36) |
| IPAQ Recreation (min/week) | 8.9 (23.9) | 26.2 (30.8) | -0.63\*\* (-0.86 to -0.40) | 9.6 (27.1) | 20.9 (28.9) | -0.40\*\* (-0.71 to -0.10) |
| IPAQ Sitting Total (hrs/week) | 60.1 (25.3) | 54.9 (42.1) | 0.15 (-0.08 to 0.38) | 58.6 24.3 | 55.4 (40.0) | 0.10 (-0.20 to 0.40) |

\*Data are mean ± standard deviation (SD); BMI = Body Mass Index; ES = Effect size difference between groups (Hedges' *g*) [48]; CI = 95% confidence interval for the measured ES; Frequencies are reported as Pearson Chi-Square. IPAQ = International Physical Activity Questionnaire [52]

\*\*significant difference between groups based on ES and CI (α≤0.05); HR = heart rate; DBP = diastolic blood pressure; SBP = systolic blood pressure

\*\*\*PROMIS Physical Function T-Score [53]

#Categories = Less than High School, High School Graduate, College Graduate, Post College

##Categories = White, Black/African American, Asian/Pacific Islander, Other (missing data included 5% for ME/CFS and 15% for controls)

###Categories = Current Fibromyalgia (FM), Irritable Bowel Syndrome (IBS), Migraine

╫Categories = Angiotensin Converting Enzyme (ACE) Inhibitor, Angiotensin Receptor (AR) Blocker, Beta Blocker, Calcium Channel 2 (CA2) Blocker.

Secondary analyses were performed on our dynamic exercise responses (i.e., 20%-100%) controlling for the presence of the most frequent and current comorbid illnesses that are commonly associated with ME/CFS (i.e., fibromyalgia (FM), irritable bowel syndrome (IBS), migraine). These analyses were conducted to determine whether comorbid illness had a substantial effect on our primary outcomes. Determining the specific impact of each comorbid illness on the cardiopulmonary responses to exercise was beyond the scope of the current investigation.

## Results

### Data quality

Of the 411 exercise tests available for data quality inspection, eight tests were excluded: 4 due to incomplete tests (less than 2 minutes of data), and 4 due to subjects withdrawn from study (reasons unknown). Of the remaining 403 tests, 363 (90%) were complete and met standardized criteria for peak effort. Of the 40 tests not meeting criteria, 35 were due to submaximal efforts (i.e. the peak HR, respiratory exchange ratios and RPE were found to be below criteria values) and 5 were due to erratic metabolic and HR data that precluded peak interpretation. Participant illness status (cases vs. controls) was unblinded after data quality assessment was completed. The test results from 16 ill controls (e.g. participants with FM only, chronic Lyme disease) were not included in subsequent analyses due to the small group size. The final analysis sample included 347 tests from 178 ME/CFS and 169 control participants. The fitness-matched subgroup was classified as 99 pairs of participants (99 ME/CFS and 99 controls) who were matched for age (±5 years) and peak $\dot{V}O_2$ (~1 ml/kg/min-1).

### Participant characteristics

Demographic and baseline variables for both the final analysis sample and the fitness-matched subsample are presented in Table 1 (additional descriptors of the group with ME/CFS are included in S1 Table). For the overall sample, participants with ME/CFS were moderately older than controls and there were small ($p<0.05$) effect size differences for weight and BMI with greater values for ME/CFS. The control group was more diverse (59% White). The fitness-matched subsample groups did not have significant or meaningful differences in any demographic variable except for race. A larger percentage of participants with ME/CFS had a current comorbid illness of FM, IBS, and/or migraine compared to controls. There were small to moderate differences for HR and blood pressure between ME/CFS and control groups in both the overall and fitness-matched samples. As expected, participants with ME/CFS demonstrated large ($p<0.05$) differences in self-reported physical function and moderate ($p<0.05$) differences in self-reported physical activity compared with controls. Meaningful differences (greater than 10-points in T-scores) were also observed in physical function via PROMIS Physical Function T-scores. However, there were small ($p>0.05$) differences for self-reported sitting-time.

### Exercise testing data

**Gas exchange threshold.** Cardiopulmonary responses at the GET are shown in Table 2. Compared to controls, participants with ME/CFS reached the GET at a similar percentage of their peak VO$_2$, but at a significantly ($p<0.05$) lower absolute $\dot{V}O_2$, VCO$_2$, $f_R$, HR, CTI and Watts, and significantly ($p<0.05$) higher $\dot{V}E/\dot{V}O_2$ and $\dot{V}E/\dot{V}CO_2$. Effect sizes ranged from small to moderate. In the fitness and age-matched subsample, significant ($p<0.05$) differences between ME/CFS and controls remained for $f_R$, $\dot{V}E/\dot{V}O_2$ and $\dot{V}E/\dot{V}CO_2$ with effect sizes in

**Table 2. Cardiopulmonary responses at the gas exchange threshold during exercise testing in ME/CFS patients and controls.**

| | Overall Exercise Study Sample | | | Fitness-Matched Subsample | | |
|---|---|---|---|---|---|---|
| | ME/CFS (n = 178) | Controls (n = 169) | ES (CI) | ME/CFS (n = 99) | Controls (n = 99) | ES (CI) |
| %peak $VO_2$ | 52.9 (11.0) | 51.3 (11.0) | 0.15 (-.06—to 0.36) | 52.8 (11.7) | 51.3 (10.9) | 0.14 (-0.14 to 0.42) |
| $\dot{V}O_2$ (ml) | 947.1 (396.7) | 1089.3 (503.6) | -0.31** (-0.53 to -0.10) | 997.5 (407.4) | 944.4 (395.7) | 0.13 (-0.15 to 0.41) |
| $\dot{V}CO_2$ (ml) | 801.6 (351.8) | 937.2 (462.8) | -0.33** (-0.54 to -0.12) | 849.2 (360.9) | 816.8 (352.1) | 0.09 (-0.19 to 0.37) |
| RER | 0.84 (0.07) | 0.86 (0.08) | -0.25 (-0.46 to 0.04) | 0.85 (0.07) | 0.87 (0.08) | -0.23 (-0.51 to 0.05) |
| $\dot{V}E$ (L/min) | 18.8 (7.1) | 22.3 (9.5) | -0.42** (-0.63 to -0.20) | 19.8 (7.4) | 20.1 (8.2) | -0.03 (-0.31 to 0.25) |
| $f_R$ (breaths/min) | 19.9 (5.2) | 22.1 (4.8) | -0.45** (-0.66 to -0.23) | 19.5 (4.9) | 21.6 (5.1) | -0.41** (-0.69 to -0.13) |
| $V_T$ (L/min) | 1.02 (0.41) | 1.03 (0.40) | -.02 (-0.24 to 0.19) | 1.10 (0.46) | 0.96 (0.35) | 0.34** (0.06 to 0.62) |
| $\dot{V}E/\dot{V}O_2$ | 25.5 (5.2) | 23.5 (3.2) | 0.47** (0.25 to 0.68) | 25.0 (4.9) | 23.6 (3.7) | 0.33** (0.04 to 0.61) |
| $\dot{V}E/\dot{V}CO_2$ | 30.4 (6.5) | 27.7 (3.4) | 0.52** (0.30 to 0.73) | 29.7 (6.2) | 27.7 (3.4) | 0.41** (0.13 to 0.69) |
| HR (beats/min) | 103.2 (17.6) | 108.7 (19.8) | -0.29** (-0.51 to -0.08) | 105.2 (17.2) | 107.2 (20.0) | -0.10 (-0.38 to 0.17) |
| $O_2$ pulse ($\dot{V}O_2$/HR) | 9.2 (3.5) | 10.0 (4.1) | -0.22 (-0.43 to -0.01) | 9.5 (3.6) | 9.0 (4.0) | 0.14 (-0.14 to 0.41) |
| Chronotropic Index | 0.92 (0.13) | 0.97 (0.15) | -0.36** (-0.57 to -0.14) | 0.94 (0.13) | 0.98 (0.17) | -0.25 (-0.67 to -0.11) |
| Watts | 56.0 (27.7) | 73.0 (35.2) | -0.54** (-0.75 to -0.32) | 59.2 (29.9) | 64.1 (28.1) | -0.17 (-0.45 to 0.11) |

*Data are mean ± standard deviation (SD); BMI = Body Mass Index; ES = Effect size difference between groups (Hedges' $g$) [48]; CI = 95% confidence interval for the measured ES; $\dot{V}O_2$ = $O_2$ consumption; $\dot{V}CO_2$ = $CO_2$ production; RER = respiratory exchange ratio; $\dot{V}E$ = ventilation; $f_R$ = breathing frequency; $V_T$ = tidal volume; $\dot{V}E/\dot{V}O_2$ = ventilatory equivalent of oxygen; $\dot{V}E/\dot{V}CO_2$ = ventilatory equivalent of $CO_2$; $O_2$ pulse = oxygen pulse.

**significant difference between groups based on ES and CI ($\alpha \leq 0.05$)

the small to moderate range. In addition, participants with ME/CFS had higher tidal volume compared with controls ($p < 0.05$).

**Ventilatory efficiency.** Measures of ventilatory efficiency and HR performance are shown in Table 3. Compared with controls, participants with ME/CFS had moderately and significantly ($p < 0.05$) lower ventilatory efficiency, as demonstrated by a higher $\dot{V}E/\dot{V}CO_{2nadir}$ and a lower OUES. They also demonstrated lower HR performance as demonstrated by lower % HRR, and % predicted max HR. In the fitness-matched sub-sample, the OUES and % predicted max HR were no longer significant ($p < 0.05$), but significant differences ($p < 0.05$) remained for the $\dot{V}E/\dot{V}CO_{2nadir}$ and %HRR.

**Peak.** Cardiopulmonary responses at peak exercise are shown in Table 4. Compared with controls, participants with ME/CFS had significantly ($p < 0.05$) lower peak $\dot{V}O_2$, $\dot{V}CO_2$, $\dot{V}E$, $f_R$, HR, $O_2$ pulse, CTI, Watts, Time, and Lactate and significantly ($p < 0.05$) higher $\dot{V}E/VO_2$,

**Table 3. CPET variables of ventilatory and heart rate performance during exercise testing in ME/CFS patients and controls.**

| | Overall Exercise Study Sample | | | Fitness-Matched Subsample | | |
|---|---|---|---|---|---|---|
| | ME/CFS (n = 178) | Controls (n = 169) | ES (CI) | ME/CFS (n = 99) | Controls (n = 99) | ES (CI) |
| $\dot{V}E/\dot{V}CO_{2nadir}$ | 27.8 (5.9) | 25.3 (3.1) | 0.51** (0.29 to 0.72) | 27.1 (5.4) | 25.4 (3.1) | 0.39** (0.10 to 0.67) |
| OUES | 1870.0 (0.67) | 2160.0 (0.78) | -0.42** (-0.63 to -0.21) | 1.98 (0.67) | 1.91 (0.74) | 0.09 (-0.19 to 0.36) |
| OUES$_{BSA}$ | 970.0 (0.30) | 1180.0 (0.39) | -0.61** (-0.82 to -0.39) | 1.03 (0.31) | 1.02 (0.35) | 0.04 (-0.24 to 0.32) |
| % HRR$_{adjusted}$ | 83.5 (15.7) | 89.8 (12.1) | -0.44** (-0.66 to -0.23) | 83.7 (14.7) | 88.3 (13.6) | -0.30** (-0.58 to -0.02) |
| % Predicted Max HR | 90.0 (9.8) | 93.3 (7.8) | -0.39** (-0.60 to -0.18) | 90.0 (9.1) | 92.3 (8.7) | -0.22 (-0.50 to 0.06) |

*Data are mean ± standard deviation; $\dot{V}E/\dot{V}CO_{2nadir}$ = the nadir for the ventilatory equivalent of $CO_2$; OUES = oxygen uptake efficiency slope; BSA = Body Surface Area [54]; HRR = heart rate reserve; ES = Effect size difference between groups (Hedges' $g$) [48]; CI = 95% confidence interval for the measured ES.

**significant difference between groups based on ES and CI ($\alpha \leq 0.05$).

**Table 4. Cardiopulmonary responses at peak exercise in ME/CFS patients and controls.**

| | Overall Exercise Study Sample | | | Fitness-Matched Subgroup | | |
|---|---|---|---|---|---|---|
| | ME/CFS (n = 178) | Controls (n = 169) | ES (CI) | ME/CFS (n = 99) | Controls (n = 99) | ES (CI) |
| Peak $\dot{V}O_2$ (ml/kg/min) | 23.4 (8.6) | 29.9 (10.9) | -0.66** (-0.88 to -0.45) | 25.2 (9.2) | 25.1 (9.0) | 0.02 (-0.19 to 0.23) |
| $\dot{V}O_2$ (ml) | 1817.3 (704.9) | 2121.2 (761.8) | -0.41** (-0.63 to -0.20) | 1915.6 (720.3) | 1865.5 (694.9) | 0.07 (-0.14 to 0.28) |
| $\dot{V}CO_2$ (ml) | 2111.0 (766.2) | 2423.9 (787.9) | -0.40** (-0.62 to -0.19) | 2210.6 (782.7) | 2159.2 (731.0) | 0.07 (-0.14 to 0.28) |
| RER | 1.18 (0.1) | 1.16 (0.08) | 0.21 (0.00 to 0.42) | 1.17 (0.09) | 1.17 (0.09) | 0.00 (-0.21 to 0.21) |
| $\dot{V}E$ (L/min) | 54.7 (21.3) | 63.0 (21.2) | -0.39** (-0.60 to -0.18) | 57.0 (22.8) | 56.3 (20.2) | 0.03 (-0.18 to 0.24) |
| $f_R$ (breaths/min) | 34.7 (10.5) | 38.9 (8.8) | -0.43** (-0.65 to -0.22) | 33.7 (10.1) | 37.5 (9.2) | -0.39** (-0.60 to -0.18) |
| $V_T$ (L/min) | 1.79 (0.59) | 1.74 (0.59) | 0.08 (-0.13 to 0.30) | 1.92 (0.64) | 1.63 (0.57) | 0.48** (0.19 to 0.76) |
| $\dot{V}E/\dot{V}O_2$ | 38.5 (9.5) | 34.0 (6.2) | 0.57** (0.35 to 0.78) | 37.4 (9.1) | 33.6 (6.7) | 0.47** (0.26 to 0.68) |
| $\dot{V}E/\dot{V}CO_2$ | 32.8 (7.4) | 29.6 4.7 | 0.51** (0.30 to 0.72) | 32.1 (7.4) | 29.1 (4.8) | 0.48** (0.27 to 0.69) |
| HR (beats/min) | 156.0 (20.2) | 166.5 (17.6) | -0.55** (-0.77 to -0.34) | 157.7 (19.1) | 161.7 (17.7) | 0.22 (-0.50 to 0.06) |
| $O_2$ pulse ($\dot{V}O_2$/HR) | 11.6 (4.2) | 12.8 (4.6) | -0.26** (-0.47 to -0.05) | 12.1 (4.2) | 11.5 (4.4) | 0.15 (-0.06 to 0.36) |
| CTI | 0.93 (0.12) | 0.96 (0.12) | -0.25** (-0.46 to -0.04) | 0.93 (0.11) | 0.95 (0.11) | -0.18 (-0.46 to 0.11) |
| Watts | 138.6 (42.3) | 163.3 (50.1) | -0.53** (-0.75 to -0.32) | 144.7 (44.6) | 146.4 (47.3) | -0.04 (-0.25 to 0.17) |
| Time (sec) | 658.3 (182.5) | 741.1 (211.0) | -0.42** (-0.63 to -0.21) | 679.8 (192.6) | 673.1 (190.1) | 0.04 (-0.24 to 0.31) |
| RPE (6–20) | 19.2 (1.0) | 18.2 (2.0) | 0.63** (0.42 to 0.85) | 19.2 (1.0) | 18.1 (2.2) | 0.64** (0.43 to 0.86) |
| Lactate (mmol/L) | 7.9 (2.5) | 8.8 (2.6) | -0.34** (-0.55 to -0.13) | 8.0 (2.38) | 8.3 (2.66) | -0.12 (-0.40 to 0.16) |

$\dot{V}O_2$ = $O_2$ consumption; $\dot{V}CO_2$ = $CO_2$ production; RER = respiratory exchange ratio; $\dot{V}E$ = ventilation; $f_R$ = breathing frequency; $V_T$ = tidal volume; $\dot{V}E/\dot{V}O_2$ = ventilatory equivalent of oxygen; $\dot{V}E/\dot{V}CO_2$ = ventilatory equivalent of $CO_2$; HR = heart rate; $O_2$ pulse = oxygen pulse; CTI = chronotropic index; RPE = rating of perceived exertion; mmol = millimoles per liter.

**significant difference between groups based on ES and CI ($\alpha \leq 0.05$).

$\dot{V}E/\dot{V}CO_2$ and RPE. In the fitness-matched sub-sample, significant ($p<0.05$) differences remained for $f_R$, $\dot{V}E/VO_2$, $\dot{V}E/VCO_2$, and RPE and ME/CFS demonstrated higher peak $V_T$.

**Dynamic exercise responses.** Responses during exercise (20–100% peak $\dot{V}O_2$) are illustrated in Fig 1A to 1D (**See S3 Data for Original Units of these responses**). Compared to controls, participants with ME/CFS demonstrated significantly lower responses for $\dot{V}E$, $f_R$, HR, $O_2$ pulse, $\dot{V}O_2$/WR and CTI and significantly higher responses for $\dot{V}E/\dot{V}O_2$, $\dot{V}E/\dot{V}CO_2$ and RPE ($p<0.05_{adjusted}$). For the fitness-matched subgroup differences remained for $f_R$, $\dot{V}E/\dot{V}O_2$, $\dot{V}E/\dot{V}CO_2$, CTI and RPE ($p<0.05_{adjusted}$). In addition, results for the matched subgroup identified a significantly increased $V_T$ during exercise among participants with ME/CFS compared to controls ($p<0.05_{adjusted}$). There were no significant differences between controls and participants with ME/CFS for lactate responses at rest, during exercise or recovery for either the entire sample or the fitness-matched subgroup (See S1 Fig). Secondary analyses, controlling for the presence of current comorbid illness (i.e. FM, IBS, or migraine) did not substantially alter group differences for the entire sample. For the fitness-matched subgroup, group differences for $\dot{V}E/\dot{V}O_2$ were no longer significant ($p>0.05$).

## Discussion

The aims of this large-scale multi-site exercise study were to determine the cardiopulmonary, metabolic, and perceptual responses to maximal exercise in people with ME/CFS by examining measures of ventilatory efficiency and cardiovascular performance and directly matching for aerobic fitness. For the entire study sample, exercise responses among those with ME/CFS were characterized by reduced oxygen uptake and HR performance, inefficient ventilation,

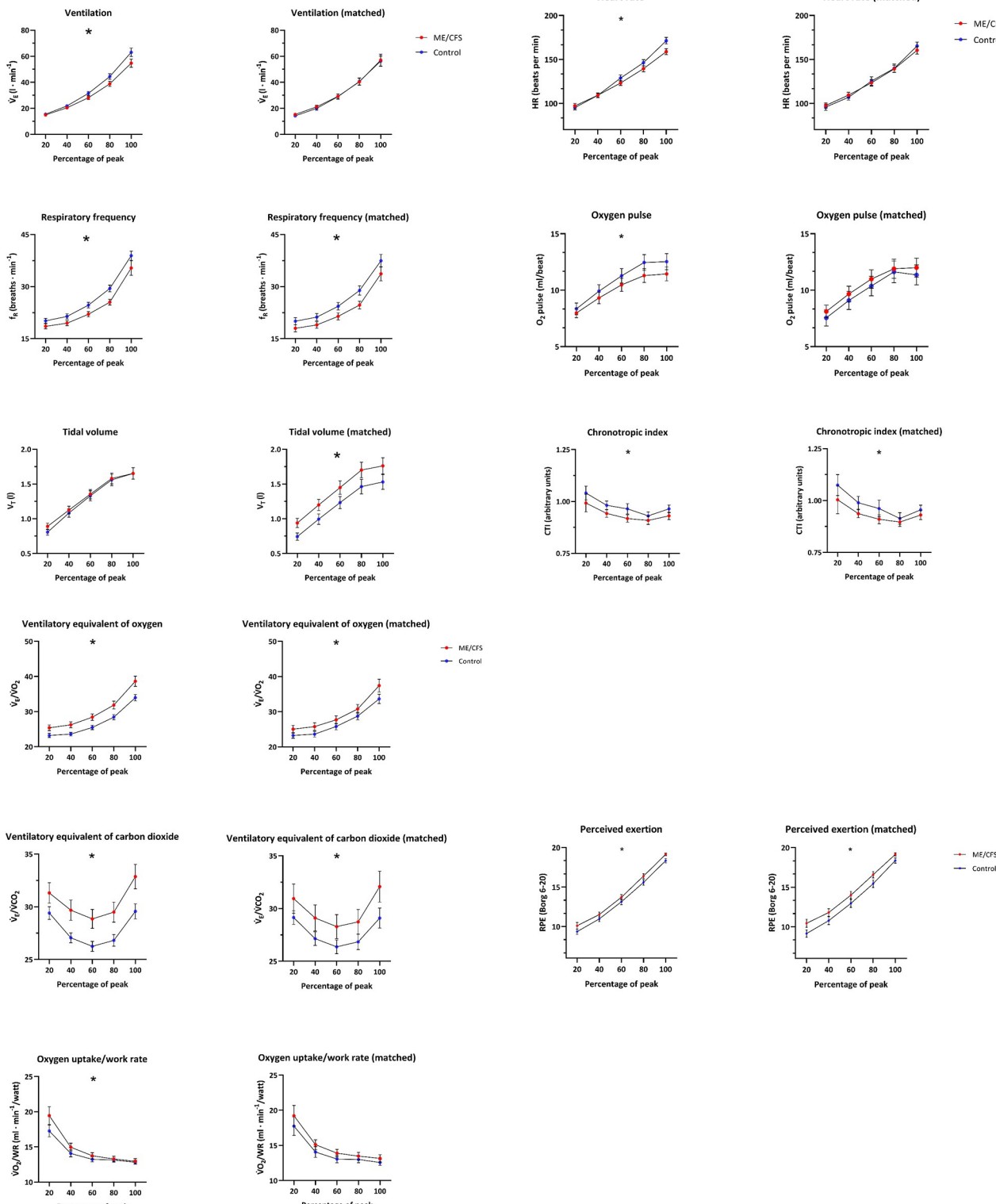

**Fig 1. Mean (95% CI) cardiopulmonary exercise testing values for participants with ME/CFS and otherwise healthy controls.** Plots in the left column show values for the full study sample (ME/CFS = 178; Controls = 169) and plots in the right column show values for the fitness-matched subgroup (ME/CFS = 99; Controls = 99). Data are expressed as 20, 40, 60, 80, and 100% of peak oxygen uptake. Significant findings from the linear mixed effects models of the entire exercise response are denoted with *. a. Ventilatory parameters—ventilation ($\dot{V}E$), respiratory frequency ($f_R$), and tidal volume ($V_T$). b. Heart rate parameters—heart rate (HR), oxygen pulse ($\dot{V}O_2/HR$), and chronotropic index (CTI). c. Efficiency related

parameters—ventilatory equivalent for oxygen ($\dot{V}E/\dot{V}O_2$), ventilatory equivalent for oxygen ($\dot{V}E/\dot{V}CO_2$), and oxygen uptake/work rate ($\dot{V}O_2$/WR).
d. Ratings of perceived exertion (RPE) on the Borg 6–20 scale.

and elevated perception of effort in comparison with controls. Many of these differences, particularly those involving cardiometabolic responses, were eliminated when matching for aerobic fitness. However, important differences in ventilatory efficiency, breathing patterns, and RPE remained. These results show that cardiopulmonary responses to exercise among those with ME/CFS are characterized by inefficient exercise ventilation.

Our results for the overall sample largely replicate what has been reported in the majority of previous studies—that people with ME/CFS are less fit than otherwise healthy controls [1, 5, 6, 25–29]. Differences in peak oxygen consumption averaged 6.5 ml/kg/min, a value that exceeds the minimal detectable change in adults with ME/CFS (5.1 ml/kg/min; c.f. Table 4 [55]) and is similar to the 5.2 ml/kg/min difference reported in a recent meta-analysis of peak aerobic capacity in people with ME/CFS [34]. To statistically control for differences in aerobic capacity, we expressed the data relative to each individual's peak oxygen consumption and only included exercise tests that met our a priori standardized criteria for peak effort. Even with these adjustments, we observed reduced ventilation and HR responses and lower ventilatory efficiency indices (e.g., higher $\dot{V}E/\dot{V}CO_2$ & $\dot{V}E/\dot{V}O_2$ and lower OUES & %HRR) for participants with ME/CFS compared to controls. The differences occurred throughout exercise, including at the GET and peak indices.

Importantly, when we performed more rigorous matching for fitness (and age), many of the group differences were eliminated including $\dot{V}E$, HR and indices of oxygen delivery such as the $O_2$ pulse, OUES, $\dot{V}O_2$/WR. These results extend upon Cook et al. [56] and indicate that many of the cardiopulmonary differences that have been reported in previous studies are explained by differences in aerobic fitness, and consequently exercise time, and are not pathophysiologic characteristics of ME/CFS.

Despite fitness matching, important and novel differences remained. Among those with ME/CFS, responses differed from controls for several ventilatory measures including $\dot{V}E/\dot{V}O_2$, $\dot{V}E/\dot{V}CO_2$, $f_R$, and $V_T$. These results suggest disease specific factors affecting cardiopulmonary responses to exercise in ME/CFS principally involving reduced ventilatory efficiency. Further determining the pathophysiological significance of these results will require testing whether cardiopulmonary responses to exercise are predictive of disease outcomes post-exercise (i.e., PEM). Moreover, determining the clinical meaningfulness of the differences observed here would require longitudinally examining whether changes in these outcomes are associated with improvements or decrements in clinically relevant outcomes (e.g., symptom severity), an approach that is not possible in this cross-sectional analysis. Further, while the convention of $\geq 0.5$ SD is sometimes used as a metric to gauge clinical significance, it is also recommended that the practical interpretation of effect size magnitude be relativized to a particular area of study [57]. In the context of CPET research involving people with ME/CFS, this study is one of the largest of its kind, especially in terms of those which controlled for aerobic fitness. Therefore, the magnitude of the differences observed here should be viewed as reference values which future CPET studies involving people with ME/CFS can begin to make inferences about clinically meaningful changes. We acknowledge that discussion of potential mechanistic explanations for the differences observed in this study should be tempered by these considerations.

Although oxygen appears able to effectively reach the periphery and be utilized, our results suggest that individuals with ME/CFS do so in an inefficient manner. These gas-exchange

inefficiencies are reflected on CPET primarily by increased $\dot{V}E/\dot{V}CO_2$ nadir and $\dot{V}E/\dot{V}O_2$ at peak exercise. The elevated $\dot{V}E/\dot{V}CO_2$ nadir reflects mismatch between ventilation and perfusion to active skeletal muscle; the peak $\dot{V}E/\dot{V}O_2$ data suggest a higher ventilatory cost of oxygen uptake perhaps due to poor extraction from skeletal muscle. Inefficient exercise ventilation is, however, non-specific and may reflect pulmonary, cardiac, and/or metabolic mechanisms. We do not believe there is evidence to support a pulmonary or HR mechanism (discussed below) but hypothesize these inefficiencies may be attributable to metabolic features of ME/CFS [40, 58–61]. Abnormalities in cellular metabolism were suggested by a retrospective observational study using finger plethysmography that demonstrated impaired oxygen extraction from exercising muscle at both the GET and peak effort among participants with ME/CFS despite normal and similar values for stroke and HR indexes [40]. In addition, a recent report of reduced deformability of erythrocytes from individuals with ME/CFS compared with controls may provide a contributing mechanism for metabolic change [62]. Reduced erythrocyte deformability, or stiffness, likely impairs microvascular perfusion and tissue oxygenation that could manifest in metabolic changes and exercise intolerance.

Although we observed gas-exchange inefficiencies in participants with ME/CFS, this does not appear attributable to a hyperventilatory response as $\dot{V}E$ was similar to fitness-matched controls across exercise intensities. It is important to note that a given $\dot{V}E$ can be accomplished by varying both the rate ($f_R$) and depth ($V_T$) of breathing. However, independent changes in $f_R$ and $V_T$ are often overlooked when interpreting CPET despite evidence of their differential control [63]. In our fitness-matched subgroup, we observed a unique breathing strategy amongst individuals with ME/CFS characterized by a slower rate and greater depth–i.e., reduced $f_R$ and increased $V_T$. This effect was greatest for $f_R$ which was observed in both our entire sample and matched subgroup (Fig 1B/Tables 2 & 4). We have reported this same inefficient ventilatory strategy in a small group of veterans with Gulf War Illness who share substantial symptom overlap with ME/CFS [64]. For those with GWI, we speculated that exercise ventilation characterized by reduced $f_R$ and increased $V_T$ may be a learned strategy to reduce symptom exacerbation or PEM. The same interpretation may hold for the present study and warrants additional investigation. Additional research into underlying mechanisms, such as mitochondrial function, is also needed to further understand the observed gas-exchange inefficiencies.

In this study, the largest difference observed during exercise was for RPE (Fig 1D/Table 4). Participants with ME/CFS rated exercise as requiring more effort throughout the test and these differences were maintained after matching for aerobic fitness. These results are consistent with a recent meta-analysis of 37 studies (involving 1016 with ME/CFS and 686 healthy controls) reporting large effect-size (d = 0.85) differences in RPE [65]. Based on the preponderance of data, it can be concluded that people with ME/CFS perceive exercise as requiring more effort than otherwise healthy people. The mechanisms for elevated RPE in ME/CFS are not fully understood but may result from the inefficient breathing patterns that we observed. Ventilation during moderate-to-high intensity exercise, is considered one of the strongest central signals for RPE [66]. Moreover, data from other illnesses and transcranial magnetic stimulation studies [67] suggest that the fatigue and pain associated with ME/CFS influence the perception of effort through interactions with skeletal and respiratory muscle signaling. Because exercise has consistently been shown to require greater effort for ME/CFS, even when matched on aerobic fitness, RPE should be considered when prescribing exercise in ME/CFS to aid in accurate prescriptions.

From a HR perspective, we saw little evidence for overt chronotropic incompetence in this large sample of participants with ME/CFS. Chronotropy during exercise is generally

determined by a combination of parasympathetic withdrawal and direct sympathetic stimulation of cardiac accelerator nerves within the heart and is dependent on the intensity of the exercise stimulus. Although the comparison between participants with ME/CFS and controls in the full exercise study sample indicated lower HR and oxygen pulse responses throughout exercise, these differences were eliminated in the fitness matched subgroup. Further, none of our clinical measures of exercise chronotropy met criteria for chronotropic incompetence [46]. On average, participants with ME/CFS achieved $\geq$ 90% of predicted peak HR, $\geq$ 80% of HRR, and had CTI $\geq$ .90 throughout exercise. These results differ substantially from several previous ME/CFS exercise studies. De Becker and colleagues [5], in a large cohort of ME/CFS (n = 427) and controls (n = 204), reported that only 37% of participants with ME/CFS achieved both a respiratory quotient of 1.0 and 85% of maximal HR compared to 80% of controls. They concluded that "reaching the age-predicted target heart rate seemed to be a limiting factor of the patients with CFS in achieving maximal effort". However, the criteria for maximal effort in their study were not standard (i.e., RER > 1.0) and different exercise work rates were used for ME/CFS (10 W/min) and controls (30 W every 3 min). Similarly, Montague et al. [41] reported normal resting HR function, but slow acceleration of the HR response during exercise. A recent meta-analysis also found large (d = -1.37) peak HR differences between ME/CFS and controls [8], however, most of the included studies did not match for aerobic fitness, did not express the data relative to peak exercise capacity, and few applied standardized criteria for peak effort determination. We conclude that the reduced heart rate responses observed in many exercise studies of ME/CFS are likely a methodological artifact and do not demonstrate chronotropic incompetence.

The primary limitation of this study is the indirect nature of CPET and thus our interpretation of the data as representing preserved oxygen delivery, but impaired utilization. Studies that include more direct measures of oxygen delivery and utilization (e.g. invasive CPET [58, 68]), and include additional measures of ventilatory mechanics and mitochondrial function, are needed to further test the mechanisms of ventilatory inefficiency that we observed. The choice of a single ramp rate for all participants resulted a range of exercise durations and differences based on fitness. However, exercise duration differences were eliminated for the fitness-matched subgroup. Future research employing individualized work-rate increments will be important towards replicating and extending the present findings and further determining aerobic fitness in ME/CFS. To our knowledge, these studies have not been conducted. The ME/CFS group was predominantly white while the control group was more diverse. Although covariation for race did not substantially alter our results, future research determining the impact of race on cardiopulmonary responses to exercise in ME/CFS is warranted. There is a paucity of data that directly compares exercise responses as a function of race, however limited data suggest that African Americans have reduced cardiorespiratory fitness and enhanced blood pressure response to exercise [69, 70]. We only determined whether the presence of comorbid illness influenced the dynamic responses to exercise, not how the specific comorbid illnesses affected the cardiopulmonary system. This would have required repeating analyses for each of the FM, IBS and migraine subgroups and thus was considered well beyond the scope of the current study. Future work aimed at determining the differential effects of these comorbidities is needed. The small sample size of the illness comparison group also precluded tests that would have helped determine whether the observed results were unique to ME/CFS or a shared pathophysiology among chronic multisymptom illnesses. A common issue of exercise research in ME/CFS is that only those who are able to exercise, volunteer for such studies. This is not unique to the current investigation but does limit the degree to which the result might generalize to a more severely affected person with ME/CFS. It is notable that our measure of physical function (PROMIS Physical Function Score) did not substantially change for

the overall compared to fitness-matched groups. There were also notable strengths to this study including a large sample from multiple clinics, adherence to standardized exercise protocols and criteria for volitional effort, independent and blind assessment of cardiopulmonary data, and the ability to match groups on age and fitness.

## Conclusion

In general, the acute exercise capacity of this cohort of people with ME/CFS was in the low-to-normal range, when considering their GET and peak aerobic capacity values. However, these data do not provide a complete functional picture of the cardiopulmonary system in ME/CFS. Ventilatory efficiency was found to be low in those with ME/CFS and significantly worse than controls. The observed responses likely reflect adequate oxygen delivery but inadequate oxygen utilization and are suggestive of disease specific adaptations that may be of pathophysiological significance but require more research. These data also highlight the importance of distinguishing fitness effects from those that are primary to the disease. By closely matching our groups on aerobic capacity/exercise time and age, many group differences were eliminated. Importantly, our data suggest that chronotropic incompetence was not present among this large sample of participants with ME/CFS.

When considering physical activity for people with ME/CFS, clinicians face the challenge of helping patients avoid the negative effects of acute exercise (e.g., symptom exacerbation) [71, 72], while moving them towards experiencing the health benefits associated with a more physically active lifestyle [73]. A logical approach is to develop exercise prescriptions which strike a balance between minimizing symptom exacerbation and maximizing function, however, there is limited information on the intensity threshold at which this ideal balance occurs or guidance on how to establish this threshold for individual patients. It is noteworthy that in other patient care settings for which a substantial literature on exercise prescription already exists, ramped incremental CPET is considered the gold standard for physiologically comprehensive exercise intensity assessment and prescription [74]. Given that over 90% of the present sample was able to provide a valid peak effort during CPET, we conclude that there is sufficient precedent for future work testing whether CPET guided exercise prescription can help address the unique physical activity challenges experienced by people with ME/CFS. Further, we believe that these data will support current recommendations to practitioners to encourage patients with ME/CFS to maintain tolerated levels of activity, to increase activity with caution, and make adjustments to avoid post-exertional malaise.

## Supporting information

**S1 Data. Group comaprisons adjusting for cardioactive medications.**
(PDF)

**S2 Data. Mixed effects models adjusting for cardioactive medication.**
(PDF)

**S3 Data. Original Units for dynamic responses to exercise.**
(PDF)

**S1 Fig. Lactate responses to exercise.**
(DOCX)

**S1 Table. Characteristic of participants with ME/CFS–overall functioning and symptom status.**
(DOCX)

## Acknowledgments

We would like to thank the participants for their efforts and volunteerism.

**MACAM Study Group:** The Multi-Site Clinical Assessment of Myalgic Encephalomyelitis/ Chronic Fatigue Syndrome (MCAM) Study Group included the following: Centers for Disease Control and Prevention/ Division of High-Consequence Pathogens and Pathology/Chronic Viral Diseases Branch, Atlanta, Georgia: Elizabeth Unger (principal investigator), Jin-Mann Sally Lin (co- principal investigator), Yang Chen, Monica Cornelius, Irina Dimulescu, Elizabeth Fall, Britany Helton, Maung Khin, Mangalathu Rajeevan; Bateman Horne Center, Salt Lake City, Utah: Lucinda Bateman (principal investigator), Jennifer Bland, Patricia Jeys, and Veronica Parkinson; Hunter-Hopkins Center, Charlotte, North Carolina: Charles Lapp (principal investigator) and Wendy Springs; Institute for Neuro Immune Medicine, Miami, Florida: Nancy Klimas (principal investigator), Elizabeth Balbin, Jeffry Cournoyer, Melissa Fernandez, Shuntea Parnell, and Precious Leaks-Gutierrez; Mount Sinai Beth Israel, New York, New York: Benjamin Natelson (principal investigator), Michelle Blate, Gudrun Lange, Sarah Khan, and Diana Vu; Open Medicine Clinic, Mountain View, California: Andreas Kogelnik (principal investigator), Joan Danver, David Kaufman, Macy Pa, Catt Phan, and Sophia Taleghani; Richard N. Podell Medical, Summit, New Jersey: Richard N. Podell (principal investigator), Trisha Fitzpatrick, and Beverly Licata; and Sierra Internal Medicine, Incline Village, Nevada: Daniel Peterson (principal investigator), Elena Lascu, Gunnar Gottschalk, Marco Maynard, and Janet Smith.

**Disclaimer:** The findings and conclusions in this report are those of the authors and do not necessarily represent the official position of the Centers for Disease Control and Prevention.

## Author Contributions

**Conceptualization:** Dane B. Cook, Stephanie VanRiper, Ryan J. Dougherty, Jacob B. Lindheimer, Michael J. Falvo, Yang Chen, Jin-Mann S. Lin, Elizabeth R. Unger.

**Data curation:** Dane B. Cook, Stephanie VanRiper, Ryan J. Dougherty, Jacob B. Lindheimer, Michael J. Falvo, Yang Chen, Jin-Mann S. Lin, Elizabeth R. Unger.

**Formal analysis:** Dane B. Cook, Stephanie VanRiper, Ryan J. Dougherty, Jacob B. Lindheimer, Michael J. Falvo, Yang Chen, Jin-Mann S. Lin, Elizabeth R. Unger.

**Funding acquisition:** Dane B. Cook, Jin-Mann S. Lin, Elizabeth R. Unger.

**Investigation:** Jin-Mann S. Lin, Elizabeth R. Unger.

**Methodology:** Dane B. Cook, Jin-Mann S. Lin, Elizabeth R. Unger.

**Project administration:** Jin-Mann S. Lin, Elizabeth R. Unger.

**Supervision:** Elizabeth R. Unger.

**Writing – original draft:** Dane B. Cook.

**Writing – review & editing:** Stephanie VanRiper, Ryan J. Dougherty, Jacob B. Lindheimer, Michael J. Falvo, Yang Chen, Jin-Mann S. Lin, Elizabeth R. Unger.

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
