## [Decision Letter · Decision Letter 0]

2 Nov 2021

PONE-D-21-16510Cardiopulmonary, metabolic and perceptual responses during exercise in Myalgic Encephalomyelitis/Chronic Fatigue Syndrome (ME/CFS): A Multi-site Clinical Assessment of ME/CFS (MCAM) sub-studyPLOS ONE

Dear Dr. Cook,

Thank you for submitting your manuscript to PLOS ONE. After careful consideration, we feel that it has merit but does not fully meet PLOS ONE’s publication criteria as it currently stands. Therefore, we invite you to submit a revised version of the manuscript that addresses the points raised during the review process.

You will see that the reviewers' comments are quite contrasted. As you are probably aware, your paper can still be rejected if adequate changed and responses to reviewers are not adequate.

We look forward to receiving your revised manuscript.

Kind regards,

Guillaume Y. Millet, PhD

Academic Editor

PLOS ONE

Journal Requirements:

3. One of the noted authors is a group or consortium 'The MCAM Study Group'. In addition to naming the author group, please list the individual authors and affiliations within this group in the acknowledgments section of your manuscript. Please also indicate clearly a lead author for this group along with a contact email address.

Reviewers' comments:

Reviewer's Responses to Questions

**Comments to the Author**

1. Is the manuscript technically sound, and do the data support the conclusions?

Reviewer #1: Partly

Reviewer #2: Yes

Reviewer #3: Yes

2. Has the statistical analysis been performed appropriately and rigorously? 

Reviewer #1: I Don't Know

Reviewer #2: Yes

Reviewer #3: No

3. Have the authors made all data underlying the findings in their manuscript fully available?

Reviewer #1: Yes

Reviewer #2: Yes

Reviewer #3: No

4. Is the manuscript presented in an intelligible fashion and written in standard English?

Reviewer #1: Yes

Reviewer #2: Yes

Reviewer #3: Yes

5. Review Comments to the Author

Reviewer #1: This study examined cardiorespiratory responses to exercise in a group with Myalgic

Encephalomyelitis/Chronic Fatigue Syndrome (ME/CFS) compared to a control group. Although I value the amount of work put into this manuscript, I have some major concerns as described in detail below. As a general comment, I believe that the methodology used in this study does not allow to gain the mechanistic insights that are proposed in the manuscript. This leads to an overly speculative discussion that is not supported by the data the authors presented.

Introduction

Lines 67: “has been” or “is”?

Lines 69-70: Are 12 references needed to support this statement? I know it is important to recognize other people’s work. However, this seems excessive.

Lines 74-75: What do you mean by “earlier ventilatory threshold onset”? Are you indicating a lower percent value or earlier in the strict sense of time? I am asking because the actual duration to achieve a given threshold might be not that relevant as this is protocol and fitness level dependent.

Line 77: Well, I guess it all depends on the characteristics of the “control” group.

Line 85: Again, the authors should specify what they mean by “early”.

Line 93: What are the authors referring to by “cardiac acceleration”? Is this the kinetics of HR and/or cardiac output?

Line 101: Expressing as a percent of peak capacity has pros and cons. It should be considered that functional tasks require a given absolute metabolic rate, rather than a percent of max.

Methods

Line 139: Matching for sex and age is important. However, I think that matching for physical activity levels would have been very important. It is true that age might contribute to fitness level, but even within the same age group, activity levels will play a key role in maximal a submaximal responses to exercise testing. NOTE: after reading the full manuscript, I see that matching to fitness level was performed. This should be explained here in my view.

Line 149: It should read “principal” not “principle”.

Line 157: I know that you had very limited exclusion criteria. However, smoking is certainly something that will affect performance. Additionally, should you also match groups by the number of smokers?

Lines 167-168: Technically speaking, this is not a warm-up as pedaling for 1 min against no external resistance is not enough (neither in intensity nor in duration) to produce a warm-up effect. This should be called a baseline measure. Unfortunately, although baseline measures are quite useful to stabilize the signal, a 1-min baseline is too short for that purpose. Finally, using 0 W is not the best choice as you do not really know the PO associated with baseline. I mean, the external workload is 0 W, but there is some power that needs to be generated to move the legs (which is ~15 W depending on the size of the person). This is why baseline measures often start at 20 W, so that changes in PO can be measurable.

Lines 179-180: Why minute 2 of the exercise? I guess it is fine because even a very unfit person should still be in the moderate intensity domain at 30 W. However, using time is tricky for this type of measurements. Also, it is odd to have a measure of lactate at peak exercise (I assume immediately after exercise cessation), but not at 1 min. By minute 3, lactate might already be going down, and the highest concentration might not have been caught at peak exercise.

Line 182: Actually, the V-Slope method defines the gas exchange threshold (GET). The ventilatory threshold (VT) requires the evaluation of ventilatory data. Additionally, if referring to the VT as synonymous of the GET, then this should be defined as VT1, as there is also the VT2, which corresponds to the respiratory compensation point (RCP). Personally, I would never determine thresholds based solely on one plot. I would rather combine plots to feel more confident in the outcome. Regardless, please use the correct terminology here and elsewhere.

Line 183: Reference #23 is not Sue et al. This worries me as it might mean that references are misplaced not only here.

Lines 183-186: Using secondary criteria has limitations that have been highlighted elsewhere (PMID: 17968581). Independently of that, some of the values that you accepted are too low in my view. For example, HR >85% of predicted HRmax is a very low mark. Even RPE greater than >17 seems not too demanding for establishing maximal efforts. Also, what do you mean with “≤150 ml with an increase in work”? Based on your methods, you increased 5 W every 20 s. Are you implying every 20 s or every 1 min for this increase in VO2? Regardless, given that the increase in PO even per min is only 15 W, the expected increase in VO2 responses might be lower than 150 mL/min even if a plateau is not reached. This might be especially true towards the end of the test (PMID: 31580218). Thus, the criteria that you are proposing is quite unlikely to establish a maximal effort (which you should aim for, independently of whether you call it VO2peak or VO2max).

Line 188: I wonder how oxygen extraction can be derived from these outcomes.

Line 190: Based on what I discussed above with the 0 W baseline, you cannot calculate the function gain (i.e., ΔVO2/ΔWR) with any level of confidence as you do not know the actual baseline PO. Again, the external resistance was set to 0 W, but moving the legs against no external resistances required an unknown PO. Additionally, after examining the results in the graphs, the functional gain values do not make much sense. How can people be more efficient at greater intensities of exercise when the gain in the heavy and severe domains is progressively greater than that in the moderate domain? This has to do with limitations of using the ramp model to understand efficiency (PMID: 31580218).

Lines 194-201: All these estimations make little sense as they are based on too many assumptions.

Results

Lines 275-276: OK, this is good to know. I think you could provide with more information on this in the previous section when you refer to this point.

Line 281: I think it should be “body mass” instead of “weight” (here and elsewhere, including tables).

Lines 304-362: Aside from some small differences still present in the matched groups, what these data show (not surprisingly) is that fitness level modulates the responses. I would accept that someone with the condition evaluated in this study might be more likely to be less fit. However, once fitness level is matched, the condition is no longer the problem. I see that there are some minor differences still present in the groups matched by fitness level (mostly some ventilatory responses with breathing frequencies and tidal volumes), but nothing very concerning. In fact, some of the values seem very close despite the claim of significance. I guess all this will be discussed later, but I wanted to share my thoughts at this point of the process.

Discussion

Lines 386-389: First, I do not see the differences to be very large despite some being significant. Regardless, I accept that the authors need to discuss this, but I would tone it down a bit when referring to “important differences”. Most importantly, the statement that “those with ME/CFS are characterized by inefficient exercise ventilation suggestive of preserved oxygen delivery but poor oxygen utilization” cannot be justified by the data that the authors have presented. There is no information in this manuscript on oxygen delivery at the central or peripheral level, or anything that would imply a deficit for oxygen utilization at the mitochondrial level. Thus, these data show some differences between groups in some variables that were evaluated, but nothing that could be translated into this assertion.

Lines 390-412: OK, I understand that some of the differences need to be highlighted. However, the most important message is not that there were some differences in ventilatory patterns even when matching by fitness level, but rather that the matching strategy abolished the ME/CFS decline in fitness and other responses. Thus, it seems as if ME/CFS patients can have similar fitness as the control participants.

Line 416: I am not sure I agree with the idea of a ventilatory inefficiency. In the end, the ventilatory responses adjust to respond to muscle metabolism. Thus, different metabolic conditions within the active tissues might cause the differential responses in the ME/CFS patients. It would be interesting to know more about what was happening at that level.

Lines 419-435: This is highly speculative. Trying to imply peripheral limitations from these data seems excessive.

Lines 436-452: I still find the level of speculation too high. At least in this case, it has to do with something that was measured in this study. The first theory seems interesting as avoiding negative symptoms is something that people try to do. The second one seems less appealing to me as I am not sure there is evidence of a perfusion limitation at the alveoli level. If this was the case, would not you expect a difference in ventilatory responses even in the groups matched by fitness level?

Lines 453-468: Again, this is just mere speculation. Similar to my previous comment, would not you expect a difference in ventilation in the groups matched by fitness level if muscle fatigue in the ME/CFS group was an issue? Perhaps I am not understanding the point the authors are making. Nevertheless, I think that the authors are trying to say far more than what they can with the data that they are presenting.

Lines 475-485: I think that the increased RPE is an interesting finding. I was fine with the first part of this paragraph. However, in this section, the authors went back to the speculation. The reality is that the RPE is regulated by many different components as the authors indicated. However, none of the possible contributors to the RPE were measured in this study. Thus, there is not much to say other than “RPE was greater in ME/CFS but we do not know why”.

Lines 486-513: The cardiac data are very “soft” to try to make any mechanistic implications. This paragraph is very speculative with little connection to the data that the authors presented.

Lines 553-564: I do not see how this study can help informing practitioners on physical activity or exercise recommendations.

Reviewer #2: Thank you for submitting this manuscript for review. Please find some comments below:

Abstract

Methods - Define HR on first use rather than just stating cardiac. Also include 'rating' for RPE.

Intro

- Line 80/81 - check sentence structure

- Line 82 - This is not a comment per se, but more out of interest for the reader. It is up to you whether you choose to add this detail in. Was this study conducted in those with MW/CFS? Are there any contraindications of doing serial tests in this population? How does the results compare to healthy individuals?

Methods

- Line 212 - you mention missing data here. Please explain in the statistical analysis section how you dealt with missing data points.

- Line 247 - you state here that you did not specifically match for sex. Yet earlier on in the methods (line 139) you state that they were matched on sex and age. Please can you clarify this.

- Line 159 - I understand the need for a screening ECG prior to exercise for safety. However, it seems as though you used an ECG during the tests but only measured was HR. Why was this? If this wasn't the case, explain that. If you did use ECG just for HR,

Results

- Tables - may look neater if there is consistency with use of decimal places. Please also ensure all acronyms are detailed in the footer. I think SBP and DBP are missing from table 1but just double check in case there are any others.

Discussion

- Line 381 - You mention cardiac performance as you have done throughout. One question I have is, have you truly measured cardiac performance with only measuring HR? Is there maybe a better term which could be used for this? If you do change it, please ensure there is consistency of use throughout the manuscript.

Reviewer #3: This is an interesting and important study and the manuscript is well written. I have highlighted a few methodological and statistical concerns that I would like to see addressed.

ABSTRACT

• Define ME/CFS and MCAM at first mention

INTRODUCTION

• The introduction is well written and provides a strong rationale for the study in the context of current evidence.

• Can the authors amend the study objectives to improve clarity? For example, “characterise the exercise capacity of the MCAM cohort” could be changes to something like “compare exercise capacity in patients with ME/FCS compared to healthy controls recruited from the MCAM study”.

METHODS

Recruitment:

• Please list the major inclusion and exclusion criteria used to identify eligible participants for the MCAM study (this allows readers to read this paper without having to read the MCSM study paper too).

Pre-exercise testing:

• What objective criteria, based on HR, ECG morphology and metabolic responses, were used to “ensure it was safe to initiate exercise testing”?

Exercise testing:

• Can the authors explain and justify why an individually-calculated work rate increment wasn’t used? The work rate increment should be set so that CPET lasts 8-12 minutes. Using the same work rate increment for a group of people with heterogenous fitness levels will lead to different CPET durations, which may influence peak VO2.

• Can the authors explain and justify why the warm-up only lasted 1-minute? It is reasonably well-established that ~3 minutes is required to reach a steady state for hearty rate and ventilation (which is the main point of the warm-up) and that is reflected in CPET guidelines (e.g. doi: 10.1016/j.bja.2017.10.020).

• More details are required in the determination of the ventilatory threshold. Were ventilatory equivalents and/or end-tidal pressures used to confirm the ventilatory threshold alongside the V-slope method? How many investigators independently assessed the ventilatory threshold, and if so, how were discrepancies resolved? Were there any instances where the ventilatory threshold could not be ascertained?

• How were data artifacts in the raw CPET data objectively defined? I.e. what constituted a data artifact?

• For future reference, the authors may wish to consider averaging the data using moving average of the middle five of seven breaths. This would automatically remove [most, if not all] data artefacts.

Statistical analyses

• Can the authors make the raw data available? This would only include anonymysed CPET data (no personal information). This aligns with PLOS ONE’s policy on data availability: “Authors are required to make all data underlying the findings described fully available, without restriction, and from the time of publication”

• Can the authors also make the SPSS syntax available? Or at least a snippet/example of the SPSS syntax used? This would allow peer reviewers and readers to reproduce the results and check the veracity of the findings.

• To my knowledge, SPSS does not report Hedges g (admittedly I haven’t used SPSS for a long time). If this is the case, how were the Hedges g and corresponding 95% confidence intervals calculated?

• Furthermore, Hedges g can be calculated in a number of different ways (see https://www.frontiersin.org/articles/10.3389/fpsyg.2013.00863/full). Therefore, please could the authors report the formula they used to calculate Hedges g?

• The authors report that sensitivity analyses (i.e. excluding participants taking cardiovascular acting drugs) did not substantially change the results. However, the actual results are not presented. Please can the authors report the findings of all sensitivity analyses as supplementary information.

• In the mixed models, were the random effects intercept-only or did they incorporate random slopes?

RESULTS

• Please give reasons why the 4 participants were withdrawn from the study, or state that the reasons are unknown.

• Did participants self-report whether they adhered to the pre-testing restrictions? (No smoking for 2 hours, caffeine for 4 hours etc).

• Please report CPET duration for both groups.

• Please report comparisons in the original units of measurements, as well as Hedges g. This could be reported as supplementary information if needed. The difference in original units is often more informative than standardised effect sizes because it can be judged against the MCID or measurement error.

• The SD for ventilatory threshold as a % of peak VO2 is implausibly tiny (0.1%). I would expect substantially higher variability. Could the authors double check their data and code/syntax? In my opinion, this is an example why making data and code publicly available is so useful.

DISCUSSION

• Can the authors elucidate on the clinical meaningfulness of some their differences? For example, do the authors think the 6.5 ml/kg/min difference exceeds the MCID or measurement error?

• Please refrain from using the term “exercise efficiency” and use “ventilatory efficiency” instead. Exercise efficiency could be confused with a measure of mechanical efficiency.

• The CPET durations are reported in the discussion – please report these in the results section (as noted above). Furthermore, please perform a test to compare these values. There appears to be a difference in duration (9 vs 12 mins), which is most likely due to the inadequate individualisation of ramp rate, which is a major limitation of the data.

• As above, please note the inadequate warm-up (1 min not sufficient to reach steady state) and one-size-fits-all ramp rate as limitations to the study that may have influenced oxygen kinetics.

I hope you find my comments helpful. Feel free to email me if you would like to discuss any of them.

Sam Orange

sam.orange@newcastle.ac.uk

6. PLOS authors have the option to publish the peer review history of their article (what does this mean?). If published, this will include your full peer review and any attached files.

Reviewer #1: No

Reviewer #2: No

Reviewer #3: **Yes: **Samuel T. Orange

---

## [Author Response · Author response to Decision Letter 0]

21 Dec 2021

Please see our detailed and comprehensive response to the reviewers. We thank them for their time and effort.

---

## [Decision Letter · Decision Letter 1]

21 Jan 2022

PONE-D-21-16510R1Cardiopulmonary, metabolic and perceptual responses during exercise in Myalgic Encephalomyelitis/Chronic Fatigue Syndrome (ME/CFS): A Multi-site Clinical Assessment of ME/CFS (MCAM) sub-studyPLOS ONE

Dear Dr. Cook,

Thank you for submitting your manuscript to PLOS ONE. All three reviewers considered that you appropriately responded their comments but one of them has two points that need to be addressed before I can accept your manuscript.

We look forward to receiving your revised manuscript.

Kind regards,

Guillaume Y. Millet, PhD

Academic Editor

PLOS ONE

Reviewers' comments:

Reviewer's Responses to Questions

**Comments to the Author**

1. If the authors have adequately addressed your comments raised in a previous round of review and you feel that this manuscript is now acceptable for publication, you may indicate that here to bypass the “Comments to the Author” section, enter your conflict of interest statement in the “Confidential to Editor” section, and submit your "Accept" recommendation.

Reviewer #1: All comments have been addressed

Reviewer #2: All comments have been addressed

Reviewer #3: (No Response)

2. Is the manuscript technically sound, and do the data support the conclusions?

Reviewer #1: Partly

Reviewer #2: Yes

Reviewer #3: Yes

3. Has the statistical analysis been performed appropriately and rigorously? 

Reviewer #1: Yes

Reviewer #2: Yes

Reviewer #3: Yes

4. Have the authors made all data underlying the findings in their manuscript fully available?

Reviewer #1: No

Reviewer #2: No

Reviewer #3: No

5. Is the manuscript presented in an intelligible fashion and written in standard English?

Reviewer #1: Yes

Reviewer #2: Yes

Reviewer #3: Yes

6. Review Comments to the Author

Reviewer #1: I would like to thank the authors for addressing my comments. This is tricky review for me to complete. On the one hand, there are components of the manuscript that are fundamentally flawed. In many occasions, the authors justify their approaches by saying that "experts" or society guidelines endorse what they did. However, something that is conceptually flawed remains wrong even if there are pieces of the literature that would help justifying it. I could explain my position in detail but it would be pointless at this stage. On the other hand, the authors have made substantial changes to the manuscript and the other reviewers seemed pretty happy with the overall content. Then, I feel that it is not fair to engage in further discussion on some points that I think remain contentious but that cannot be changed anyway. Thus, I thank again the authors for their responses and I will not provide any further comments as I do not think there is any further meaningful contribution that I can make to the process.

Reviewer #2: (No Response)

Reviewer #3: Thank you for addressing the majority of my comments. Please review a couple of additional minor comments below.

#1 More details are required in the determination of the ventilatory threshold. Were ventilatory equivalents and/or end-tidal pressures used to confirm the ventilatory threshold alongside the V-slope method? How many investigators independently assessed the ventilatory threshold, and if so, how were discrepancies resolved? Were there any instances where the ventilatory threshold could not be ascertained?

Page 6 (lines 198-200): Thank you for this question. We did not assess ventilatory equivalents or other gas exchange threshold measures in addition to the V-slope method. The V-slope method was independently assessed by two investigators. Discrepancies were resolved by a third assessor (the first author). This information has been added to the Methods. There were six cases (all ME/CFS) where a V-slope could not be derived.

To address your concern, we analyzed a subset of our sample (n = 109) using the ventilatory equivalents method in comparison to our V-slope approach using the same 2 independent investigators who performed original analyses. Overall, we found strong agreement between approaches. Specifically, the GET occurred at a similar percentage of peak VO2 (V-slope: 53.1%; Ventilatory equivalents: 55.7%). Given the similarities and focus of the present manuscript, we have decided to retain our original analyses. Moreover, as we focus our analysis and interpretation primarily on the entire exercise response (i.e., relative intensity plots) – we believe this decision is justified.

Reviewer response: Thank you for addressing my comment. However, I think there was some confusion. I did not suggest that ventilatory equivalents should be used instead of the V-slope method to determine the ventilatory threshold – they should be used together, in line with guidelines (https://doi.org/10.1016/j.bja.2017.10.020). In other words, for future reference, ventilatory equivalents should be used to help confirm the ventilatory threshold determined by the V-slope method.

Please confirm how discrepancies were objectively defined. I.e. what counted as a discrepancy? For example, we have previously used a disagreement of >7.5% to objectively define a discrepancy. Please clarify here and add this information to the methods.

# 2 To my knowledge, SPSS does not report Hedges g (admittedly I haven’t used SPSS for a long time). If this is the case, how were the Hedges g and corresponding 95% confidence intervals calculated?

Hedges g were not calculated within SPSS. We calculated it separately within Excel.

Reviewer response: This is confusing for readers because the statistical analysis section literally states in the first line that the statistical analyses were conducted in SPSS. Please add something along the lines of: “Standardised effect sizes were calculated in Microsoft Excel as the mean difference between groups divided by the pooled SD, with a Hedges g correction applied to adjust for sample bias.”

I have also just noticed that you have referred to “Hedges’ d” – this does not exist. Please change to “Hedges’ g” throughout the manuscript.

7. PLOS authors have the option to publish the peer review history of their article (what does this mean?). If published, this will include your full peer review and any attached files.

Reviewer #1: No

Reviewer #2: No

Reviewer #3: **Yes: **Samuel T. Orange

---

## [Decision Letter · Decision Letter 2]

1 Mar 2022

Cardiopulmonary, metabolic and perceptual responses during exercise in Myalgic Encephalomyelitis/Chronic Fatigue Syndrome (ME/CFS): A Multi-site Clinical Assessment of ME/CFS (MCAM) sub-study

PONE-D-21-16510R2

Dear Dr. Cook,

We’re pleased to inform you that your manuscript has been judged scientifically suitable for publication and will be formally accepted for publication once it meets all outstanding technical requirements.

Kind regards,

Guillaume Y. Millet, PhD

Academic Editor

PLOS ONE

Additional Editor Comments (optional):

Reviewers' comments:

Reviewer's Responses to Questions

**Comments to the Author**

1. If the authors have adequately addressed your comments raised in a previous round of review and you feel that this manuscript is now acceptable for publication, you may indicate that here to bypass the “Comments to the Author” section, enter your conflict of interest statement in the “Confidential to Editor” section, and submit your "Accept" recommendation.

Reviewer #3: All comments have been addressed

2. Is the manuscript technically sound, and do the data support the conclusions?

Reviewer #3: Yes

3. Has the statistical analysis been performed appropriately and rigorously? 

Reviewer #3: Yes

4. Have the authors made all data underlying the findings in their manuscript fully available?

Reviewer #3: No

5. Is the manuscript presented in an intelligible fashion and written in standard English?

Reviewer #3: Yes

6. Review Comments to the Author

Reviewer #3: Thank you for addressing my comments and congratulations on your paper.

7. PLOS authors have the option to publish the peer review history of their article (what does this mean?). If published, this will include your full peer review and any attached files.

Reviewer #3: **Yes: **Dr Samuel T. Orange

---

## [Editor Report · Acceptance letter]

4 Mar 2022

PONE-D-21-16510R2 

Cardiopulmonary, metabolic, and perceptual responses during exercise in Myalgic Encephalomyelitis/Chronic Fatigue Syndrome (ME/CFS): A Multi-site Clinical Assessment of ME/CFS (MCAM) sub-study 

Dear Dr. Cook:

I'm pleased to inform you that your manuscript has been deemed suitable for publication in PLOS ONE. Congratulations! Your manuscript is now with our production department. 

Kind regards, 

on behalf of

Professor Guillaume Y. Millet 

Academic Editor

PLOS ONE